



# A stepwise GIS approach for the delineation of river valley bottom within drainage basins using a cost distance accumulation analysis

Gasper L. Sechu[1], Bertel Nilsson[2], Bo V. Iversen[1], Mette B. Greve[1], Christen D. Børgesen[1], Mogens H. Greve[1]

[1]Department of Agroecology, Aarhus University, Tjele, 8830, Denmark
[2]Department of Hydrology, Geological Survey of Denmark and Greenland (GEUS), Copenhagen, 1350, Denmark

*Correspondence to*: Gasper L. Sechu (gasper.sechu@agro.au.dk)

**Abstract.** River valley bottoms have hydrological, geomorphological, and ecological importance and are buffers for protecting the river from upland nutrient loading coming from agriculture and other sources. They are relatively flat, low-lying areas of
the terrain that are adjacent to the river and bound by increasing slopes at the transition to the uplands. These areas have under natural conditions, a groundwater table close to the soil surface. The objective of this paper is to present a stepwise GIS approach for the delineation of river valley bottom within drainage basins and use it to perform a national delineation. We developed a tool that applies a concept called cost distance accumulation with spatial data inputs consisting a river network and slope derived from a digital elevation model. We then used wetlands adjacent to rivers as a guide finding the river valley
bottom boundary from the cost distance accumulation. We present results from our tool for the whole country of Denmark carrying out a validation within three selected areas. The results reveal that the tool visually performs well and delineates both confined and unconfined river valleys within the same drainage basin. We use the most common forms of wetlands (meadow and marsh) in Denmark's river valleys known as Groundwater Dependent Ecosystems (GDE) to validate our river valley bottom delineated areas. Our delineation picks about half to two-thirds of these GDE. However, we expected this since farmers
have reclaimed Denmark's low-lying areas during the last 200 years before the first map of GDE was created. Our tool can be used as a management tool, since it can delineate an area that has been the focus of management actions to protect waterways from upland nutrient pollution.

## 1 Introduction

The rise in the availability of high quality spatial data, especially the representation of digital terrain models (DEMs), has
brought an increase in the number of Geographic Information System (GIS) professionals striving to create methods to best describe and extract different landscape features.

Delineation of valley bottom across drainage basins is becoming increasingly important due to an acceptance of the drainage basin area as the essential management unit for sustainable water and land management (Chowdary et al., 2009). The valley bottoms act as an intermediate pathway for nutrients coming from the uplands, either as surface flow, diffuse flow to




wet areas on floodplains, or directly through stream-bed connected to underlying groundwater bodies and have the potential
      to reduce nutrients thereby protecting the surrounding aquatic environment (Langhoff et al., 2006; Dahl et al., 2007).

      The Dictionary of Earth Science defines the valley floor as "*The broad, flat bottom of a valley. Also known as valley bottom or valley plain.*" (McGraw-Hill, 2003). This can be described conceptually as an area of low slopes bounded by increasing slopes at the transition to the uplands (Figure 1). Valley bottoms are landscape features with hydrological,

geomorphological, and ecological importance (Gallant and Dowling, 2003; Hynes, 1975; Nardi et al., 2006).

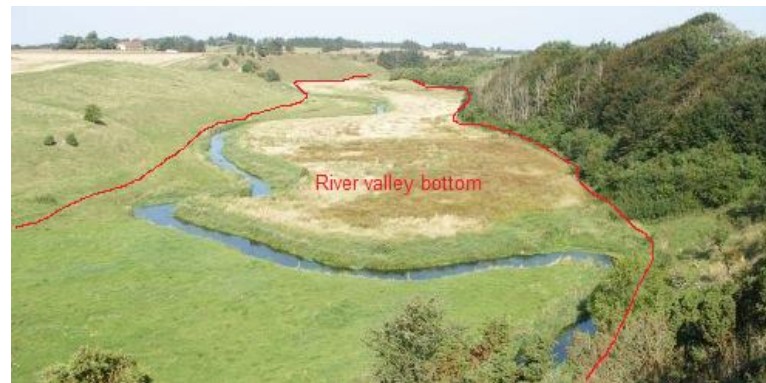

**Figure 1: Conceptualization of the river valley bottom at a section of the Villestrup stream, Denmark, showing the valley bottom as a flat to low slope area adjacent to a river and bounded by increasing slopes (Photo by the Danish Nature Agency).**


      By description, the river valley bottom delineates the river and its corresponding active floodplain (Fryirs et al., 2016; Wheaton et al., 2015). Confined, partly confined, or laterally unconfined (Figure 2) are classifications commonly used to distinguish valley bottoms (Brierley and Fryirs, 2005). Confinement of valley bottom is the percentage of natural waterway that borders a confining margin on either bank (Fryirs et al., 2016). In some situations, the valley bottom boundary can coincide

with the valley confining margin (Fryirs et al., 2016).

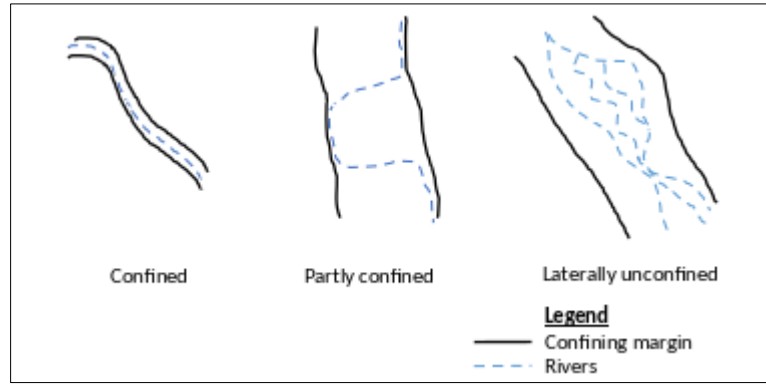

**Figure 2: Different types of valley confinement (modified from Brierley and Fryirs, 2005).**



The river valley bottom is formed by a combination of erosion, deposition, and peat formation. The valley is cut out of the landscape by erosion, either glacial or fluvial, or by tectonic processes. The formation of river valleys could fall in either of these processes or a combination depending on the situation. It can be difficult to distinguish the rate of glacial or fluvial erosion as both glaciers and rivers occupied valleys at the same time, which makes separating their contribution to valley formation a challenge (Roberts and Rood, 1984). However, some researchers have found a greater role of glacial erosion when

compared to river erosion (e.g. Clayton, 1996; Hallet et al., 1996; Kirkbride and Mathews, 1997; Montgomery, 2002), while others found a lower rate of glacial erosion or little difference between the two (e.g. Hicks et al., 1990; Summerfield and Kirkbride, 1992; Hebdon et al., 1997; Lidmar-Bergstrom, 1997). Yet, depending on the type, valley formation during the glacial time was often a combination of these different processes while during the postglacial period, the formation was mostly related to fluvial erosion and deposition.

Several GIS tools that delineate the valley bottom broadly fall into two categories: flooding or slope algorithms (Gilbert et al., 2016). The flooding algorithm works by filling water within the flat valley area finding a suitable water depth threshold that delineates the valley bottom. The slope algorithm works by finding a suitable threshold of the slope of the terrain that delineates the flat area of the valley bottom. Existing tools normally fall within these two categories (Table 1). Height Above River (HAR) uses a flooding algorithm that propagates river centerline elevations outward from the river using a distance-

weighted average and subtracts the result from the elevation (Dilts et al., 2010). Another tool that uses the flooding algorithm is River Bathymetry Toolkit (RBT), which works by detrending the DEM to remove the longitudinal slope and floods the result to investigate the extent of the stream or in this case the valley bottom (McKean et al., 2009). The Multi-resolution Valley Bottom Flatness (MRVBF) is a slope-based algorithm that uses several neighborhood calculations moving from small to large in an attempt to capture both small and large valleys, which are then combined into one single index (Gallant and

Dowling, 2003). Fluvial Corridor Toolbox is a workflow that contains several tools for extracting and classifying fluvial features. The workflow consists in part of a slope-based algorithm that extracts valley bottoms by calculating an altimetric reference plan along the river subtracting that from the original elevations to obtain a detrended elevation. This is subjected to a threshold to capture elevations that are then classified as the valley bottom (Roux et al., 2015). Finally, Valley Bottom Extraction Toolbox (V-BET) is a relatively recent tool that uses a slope-based algorithm that works as a function of the drainage

basin and scales results depending on the location within the basin (Gilbert et al., 2016).

**Table 1: Some existing valley bottom delineation tools, their data requirements, and defining algorithm.**

| Tool | Data requirements | Algorithm | Reference |
|---|---|---|---|
| Height Above River (HAR) | DEM, stream network | Flooding | Dilts et al. (2010) |
| River Bathymetry Toolkit (RBT) | DEM | Flooding | McKean et al. (2009) |
| Multi-resolution Valley Bottom Flatness (MRVBF) | DEM | Slope | Gallant and Dowling (2003) |





| Fluvial Corridor Toolbox | DEM | Slope | Roux et al. (2015) |
|---|---|---|---|
| Valley Bottom Extraction Toolbox (V-BET) | DEM, stream network | Slope | Gilbert et al. (2016) |

Tools that use the flooding algorithm such as HAR and RBT have limitations of scaling up to the entire drainage basin
since they use a single flood depth (Gilbert et al., 2016). This implies that a flooding depth that delineates the valley bottom at
the downstream side of the drainage basin results in an underestimation of the valley bottom at the upstream areas.
Correspondingly, a flooding depth that delineates the valley bottom at the upstream side would result in an overestimation at
the downstream areas (Gilbert et al., 2016). Slope-based algorithms such as MRVBF and V-BET also have scale issues
whereby slope thresholds that work for larger valleys fail to work on smaller confined ones leading to an exaggeration of the
valley bottom and vice versa (Gilbert et al., 2016). These scaling issues proved to be prevalent during our preliminary testing
of the existing tools displayed in Table 1.

Our incentive for developing a new tool for the delineation of river valley bottom was to improve on existing methods by
giving delineations that are more accurate while keeping the tool relatively simple to use. Due to their ideal location as a flat
part of the landscape, valley bottoms can create conflicts of land uses between humans and the ecosystems (Burby and French,
1981; Mount, 1995). Our approach made it possible to map whole drainage basins upscaling to a national map. We used
Denmark with an area of about 43,000 km$^2$ as an example.

The overall objective of this paper is to present a stepwise GIS approach for the delineation of river valley bottom within
drainage basins and use it to perform a national delineation. We hypothesize that based on novel GIS techniques along with
spatial inputs such as a DEM, river network, and wetland areas, we can carry out our delineation tackling issues of scaling.

## 2 Materials and methods

### 2.1 Study area

Denmark is located in Northern Europe (Figure 3) and covers an area of about 43,000 km$^2$. The country consists of the
peninsula Jutland and an archipelago of 443 named islands, the largest being Zealand and Funen. The country is comparatively
flat with a mean elevation of about 31 m above sea level and the highest point standing at about 172 m above sea level. A large
part of the terrain consists of rolling plains with sandy coastlines and large dunes located in Northern Jutland. It consists of
several streams with the largest being the Gudenå (149 km), Skjern Å (96 km), and Storå (100 km) (Ovesen et al., 2000).
Danish landscapes are a result of multiple glaciations during the Quaternary period (last 2.6 Ma) where ice covered Denmark
several times. During the 100,000 years of the last glaciation period (Weichselian glaciation), the ice advance came from the
Baltic, Norway, Sweden, and again from the Baltic and ended at the Main Stationary Line (MSL in Figure 3) (Pedersen et al.,
2012). The East Jutland ice advance (19,000 BP) came from the southeastern direction and ended at the East Jutland ice border
(E in Figure 3). This was followed by the Baelthav readvance (18,000 BP) coming from a southeastern direction going through

northwestern Zealand, the Great Belt, southern and eastern Funen, and southeastern Jutland ending at the Baelthav ice border (B in Figure 3) (Houmark-Nielsen, 2011).

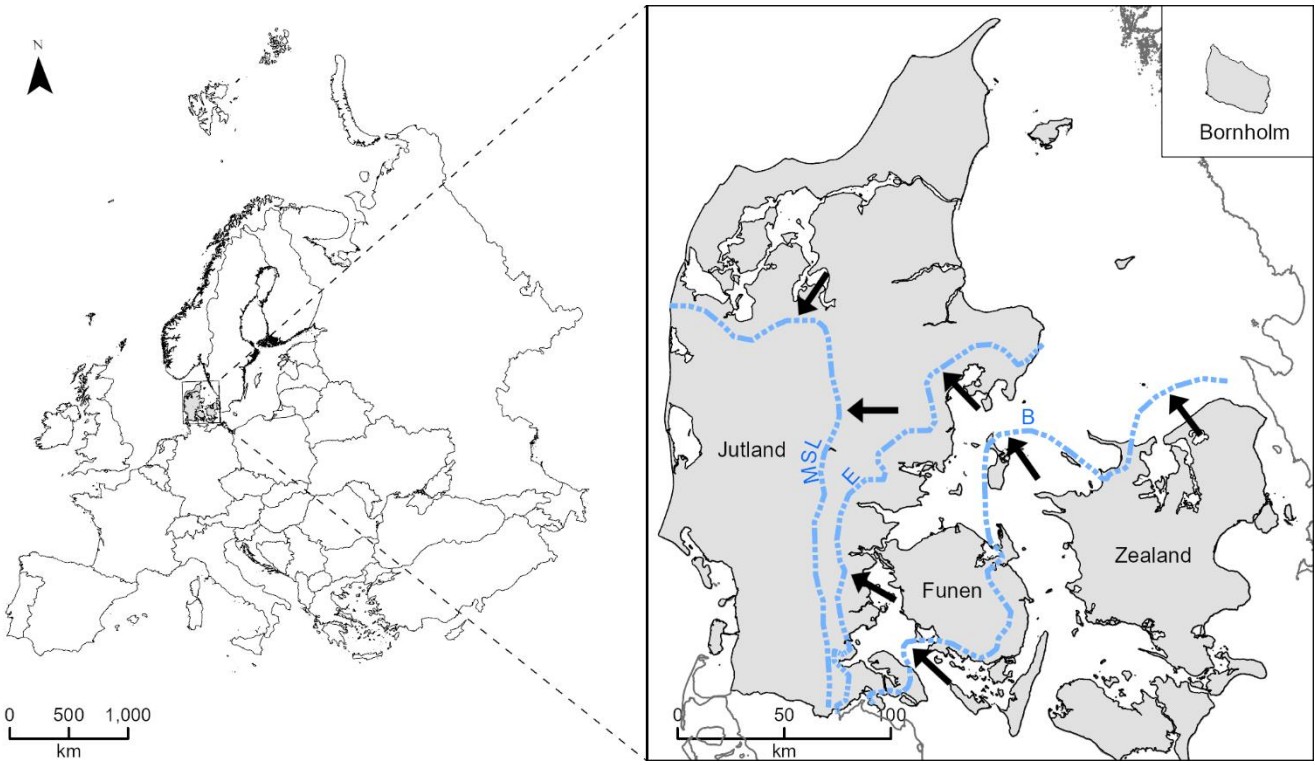


**Figure 3: The location of Denmark within Europe enlarged to show the major geomorphological regions that stem from the major ice advances within the last glaciation (arrows). MSL = Main Stationary Line, E = East Jutland ice border, B = Baelthav ice border (Houmark-Nielsen, 2011).**

**2.2 Data**

We used four datasets in the development of our river valley bottom delineation tool (Figure 4). First, a DEM of Denmark (Figure 4a). This is LIDAR data collected and processed by the Danish Agency for Data Supply and Efficiency. It has a resolution of 40 cm and can be downloaded freely ([www.sdfe.dk](www.sdfe.dk)). For the development of the tool, we resampled the LIDAR data using bilinear interpolation to a 10 m spatial resolution. We did this to remove noise in the data and reduce computation

time. Second, we used a GIS feature layer containing a river network of about 27,000 km long, spanning the entire country (Figure 4b). This dataset is prepared by the Danish Centre for Environment and Energy (DCE) at Aarhus University, Denmark, and is available in their database. The third dataset is a GIS feature layer of 142 river drainage basins (Figure 4c) also prepared by the DCE. Finally, we used a GIS feature layer of mapped historic wetland areas (Figure 4d). These are digitized from old topographical maps and cover an area of about 7,500 km$^2$ of the country (Breuning-Madsen et al., 1984). We carried out a

validation of the tool at three areas (Bjerringbro/Hvorslev, Tåstrup, and North Funen) all having GIS feature layers representing



Groundwater Dependent Ecosystems (GDE) (Figure 5). This data comes from a study by Nilsson et al. (2014) where they digitized GDE from an old Danish map (ca. 1770-1867). This is the earliest Danish detailed map (1: 5,000) representing the most undrained landscape condition before the introduction of tile drainage in the 1850s. Of the digitized nature types, meadow and marsh were used to represent the most widespread wetland types of GDE in river valleys in Denmark (CIS, 2014).


**Figure 4: Spatial data used to develop the river valley bottom delineation tool consisting of (a) Digital Elevation Model (DEM) in meters above sea level (m.a.s.l), (b) River network, (c) Drainage basins (highlighted is the Gudenå drainage basin located in the peninsula of Jutland, which we subsequently use to illustrate our delineation approach), and (d) Wetlands.**







Figure 5: Spatial data used to validate the river valley bottom delineation tool consisting of three areas: (a) Bjerringbro/Hvorslev, (b) Tåstrup, and (c) North Funen with expertly digitized GDE consisting of the nature types meadow and marsh from a study by Nilsson et al. (2014).




## 3 Methodology

### 3.1 Cost distance accumulation

We created a tool that uses a stepwise approach to delineate the river valley bottom using GIS techniques. We used the Python programming language, primarily ArcPy, a site package that is useful in customizing ESRI ArcGIS functions for geographic data analysis, management, and automation. The primary methodology that we used was cost distance accumulation. It works by quantifying surface movement based on the potential accumulative effort (known as cost) that is required to move from an

origin (source) to outward locations (destinations). The source locations are assigned cost distance accumulations of zero and the algorithm calculates cost distance accumulations outward based on resistance factors encountered with each move. These factors can be a representable magnitude of a form of resistance that the user wants to model (e.g. slope, friction, wind, etc.) The result is often used to calculate the least costly or effortless path to traverse between two locations (e.g. best hiking route, least costly route for a pipeline, wildlife corridor habitats, etc.). We calculated cost distance accumulation by using the river

centerline as the source and the slope of the terrain as the cost limiting the calculation within the boundary of drainage basins. We hypothesize that we are going to experience a greater increase in the cost distance accumulation at the boundary of the river valley bottom, which consists of increasing slopes when compared to the valley bottom. We demonstrate the algorithm in Figure 6, as well as Equations 1, 2, and 3.

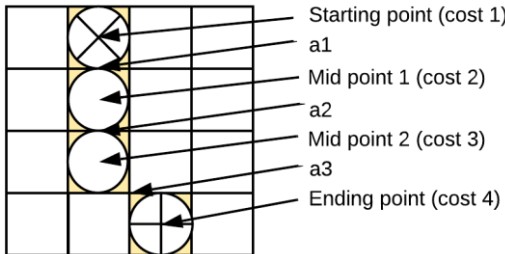


**Figure 6: Cost distance accumulation algorithm demonstrating movement from a starting point (source) to an ending point through a path that incorporates perpendicular and diagonal movement. Equations 1, 2, and 3 illustrate the computation of the costs and accumulation.**

$$a(i)_{perpendicular} = \frac{cost(i)+cost(i+1)}{2}, where \ i = 1,2,3 \dots n \tag{1}$$

$$a(i)_{diagonal} = 1.4142 \left( \frac{cost(i)+cost(i+1)}{2} \right), where \ i = 1,2,3 \dots n \tag{2}$$

$$Cost \ accumulation = \sum_{i=1}^{n} a(i), where \ i = 1,2,3 \dots n \tag{3}$$






## 3.2 A stepwise GIS approach

We processed the input DEM using a fill operation to eliminate any localized peaks and sinks before the analysis. Figure 7 illustrates the first two steps using the Gudenå drainage basin (ca. 2,700 km$^2$) located in the peninsula of Jutland (highlighted in Figure 4c). At step 1, the tool calculated a slope raster using the 10 m DEM shown in Figure 7a. It then conditioned the resulting slope raster by replacing values of zeros with small positive values resulting in a conditioned slope raster as seen in Figure 7b. We do this because the algorithm of cost distance accumulation is a multiplicative process and does not work with values of zero. At step 2, it calculated the cost distance accumulation within the drainage basin using the river network layer as the input source data and the conditioned slope raster as the input cost (Figure 7b). This resulted in a cost distance accumulation raster for the drainage basin as seen in Figure 7c. The cost distance accumulation increases depending on the magnitudes of the slopes starting from zero at the river network centerline.

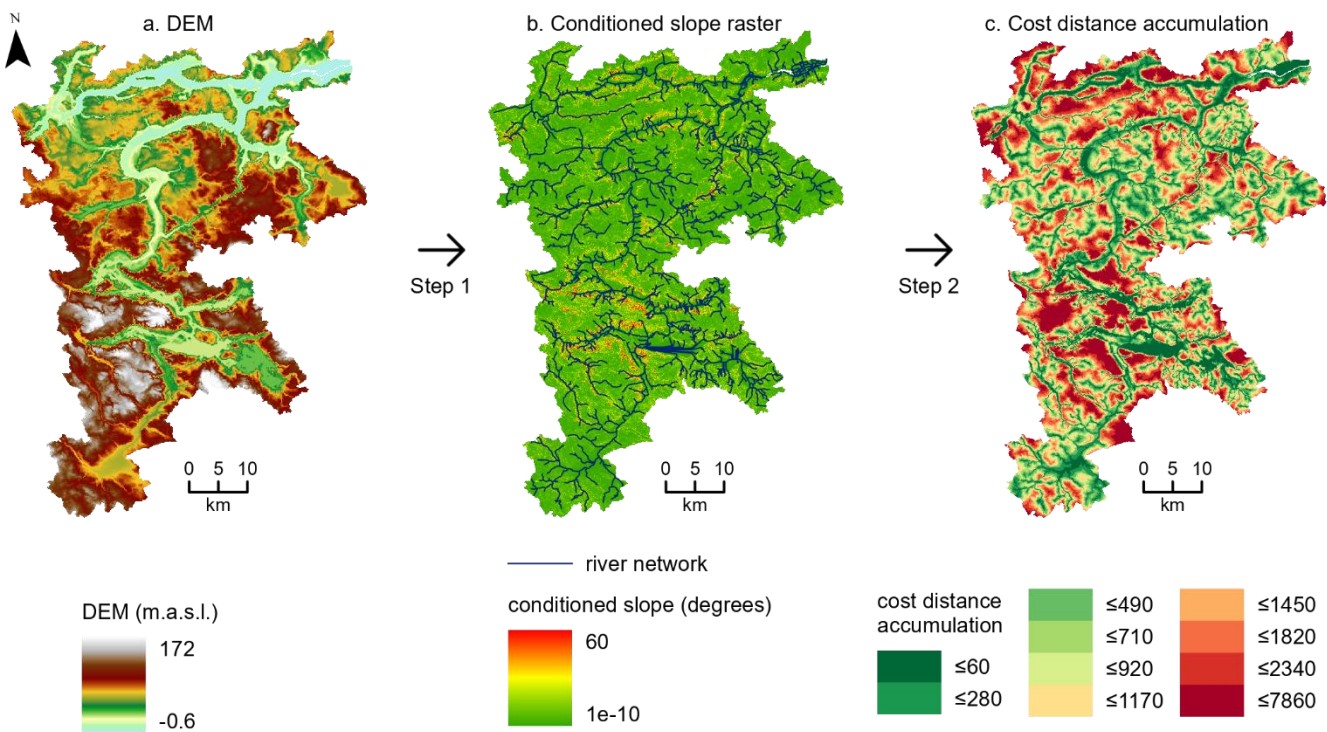

**Figure 7: Initial steps that our approach takes to delineate the river valley bottom using an example of the Gudenå drainage basin. At Step 1, using (a) the DEM as input, we calculate slope and condition it to remove values of zeros, which results in (b) a conditioned slope raster. At Step 2, we use (b) together with the river network as inputs to calculate (c) the cost distance accumulation within the drainage basin.**

We then needed a basis to extract the river valley bottom from the cost distance accumulation raster. Figure 8 illustrates the subsequent steps that lead to the final delineation of the river valley bottom for the drainage basin. We based our delineation





on calculating an estimate of the threshold cost distance accumulation that corresponds to the boundary of the river valley bottom using wetland areas that are adjacent to rivers. The tool first extracted wetland areas adjacent to rivers (Figure 8a) through a spatial analysis of proximity between input wetland and river layers. To extract a non-skewed cost distance accumulation threshold value, we filtered out zero and high outlier cost distance accumulation values located within the

extracted wetlands adjacent to rivers (Step 3 in Figure 8). The zero values correspond to values that fall directly at the river system (source) and the high values are far from the river valley bottom boundary towards the uplands. The tool extracted cost distance accumulation values ranging from greater than zero to a maximum of 500 (Figure 8b). The choice of this range comes from plotting the distribution of these values which results in a decay curve that plateaus before reaching 500 (Figure 9). The tool then calculated the mean of these values that we used as a threshold cost distance accumulation for extracting the river

valley bottom for the drainage basin (Step 4 in Figure 8). For the Gudenå drainage basin, the threshold cost distance accumulation is 90 as seen in Figure 9. The final delineated river valley bottom extracted using this threshold is shown in Figure 8c. We then created a loop that repeated this process until all Danish drainage basins were processed and finally combined into one single layer.

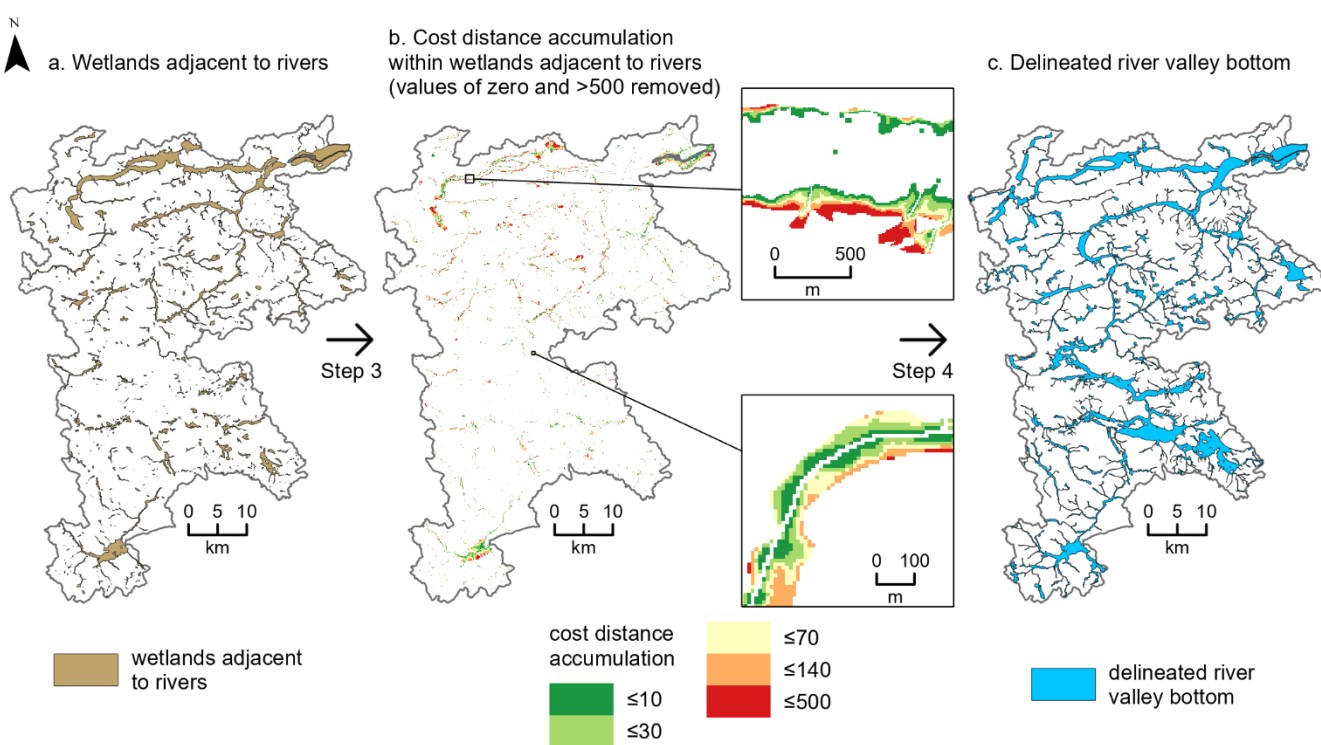


**Figure 8: Subsequent steps that our approach takes to delineate the river valley bottom. At Step 3, using (a) wetland areas adjacent to rivers as input, we extract (b) the cost distance accumulation values falling within and filter them to remove outliers (extracting only values between zero and 500). At Step 4, we use (b) to calculate the mean of the values, which we use as a basis to delineate (c) the river valley bottom from the cost distance accumulation of the drainage basin.**




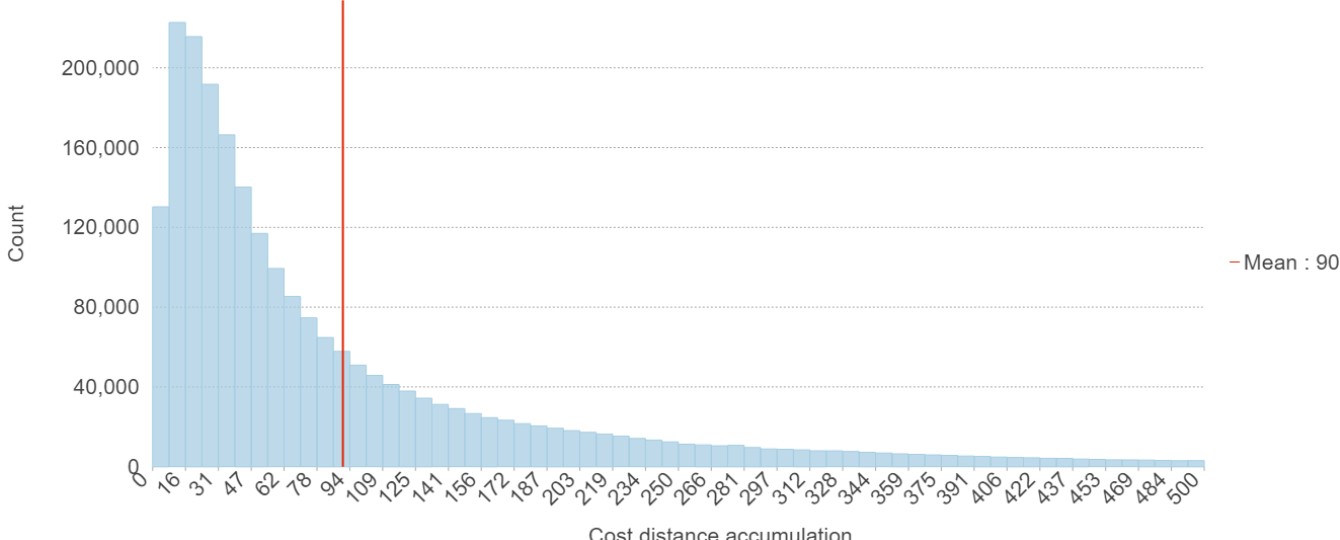

**Figure 9: Distribution of cost distance accumulation values in the range greater than zero to 500 within wetlands adjacent to rivers in the Gudenå drainage basin showing the mean value (value of 90) of cost distance accumulation (red vertical line), which we use as the threshold for our river valley bottom delineation. Count on the Y-axis represents the number of cost distance accumulation**
**raster cells within each band.**

We carried out a validation of the stepwise GIS approach by doing a percent overlap analysis using our delineated river valley bottom and GDE that are adjacent to rivers within the three validation areas (Figure 5). The analysis entailed finding an overlap between our river valley bottom delineation and the GDE. It was then converted to a percentage by dividing its area
by the area of the extent that is covered by our river valley delineation. We hypothesized that some of the delineated river valley bottom areas will contain some GDE habitat areas with meadow and marsh.

## 4 Results

### 4.1 River valley bottom map of Denmark

We ran the tool for the entire of Denmark delineating an area about 8,500 km$^2$ of the river valley bottom, which is
approximately about 20% of the country. We present the resulting map and zoom into the areas that we subsequently use for validation (Figure 10). When overlaid on a relief map generated by the 10 m DEM, the delineation visually looks good and falls on the low areas of the terrain as we expected. The tool delineates both confined and unconfined river valleys within the drainage basins. Confined river valleys can be seen as the narrow, mostly headwater river valley bottom sections, while unconfined river valleys are wider downstream sections.


**Figure 10: Delineated river valley bottom map of Denmark zoomed into the three validation areas. Confined headwater valleys have small widths whereas unconfined downstream valleys have broader widths.**





## 4.2 Validation

Our validation explored the area that GDEs close to rivers occupy within the delineated river valley bottom. This resulted in an overlap area between GDE and delineated river valley bottom that we represent as a percentage of the river valley bottom extent area, which is the area that covers the overlap and areas outside the overlap. We calculated this for the areas of Bjerringbro/Hvorslev, Tåstrup, and North Funen (Figure 11). We also present a summary of the validation results in Table 2. These percent overlaps can be translated as the approximate amount of area within our river valley bottom delineation that is wet. This implies that the larger the overlap, the better our tool can predict the river valley bottom since most of these areas are wet and/or have the groundwater table close to the surface.

**Table 2: Validation results showing the valley bottom and GDE intersection area, valley bottom extent, and overlap.**

| Area | Valley bottom and GDE intersection (km²) | Valley bottom extent (km²) | Overlap (%) |
|---|---|---|---|
| Bjerringbro/Hvorslev | 5.2 | 9.5 | 55 |
| Tåstrup | 10.6 | 15.7 | 67 |
| North Funen | 24.2 | 43.9 | 55 |





**Figure 11: Validation of the developed river valley bottom delineation tool showing the areas of overlap between the delineated river valley bottom and GDE, and the extent to which the river valley bottom covers for the three validation areas.**






## 5 Discussion

Our tool delineates the river valley bottom using the slope as the determining factor for the cost distance accumulation algorithm. This gives an advantage when scaling, since a flat area will increase the cost distance accumulation at a slower pace while a steep area increases it at a faster pace. This solves the issue of scaling, since confined headwater valleys will increase

the cost distance accumulation rapidly at a short distance away from the river centerline whereas unconfined valleys will increase the cost distance accumulation at a slower pace. The net result is that the threshold cost distance accumulation value for capturing the river valley bottom boundary is at relative distances for different types of valleys. This means that the threshold value at confined headwater valleys close to the river will be achieved at roughly the same cost distance accumulation as that of the unconfined valley that is further downstream. This can be seen in the results in Figure 10 where confined

headwater valleys and unconfined downstream valleys are both automatically captured within the different areas. The wetlands input gives a further guide to finding the threshold cost distance accumulation value through working out a mean of the values found within the flat valley area, which in turn gives a more accurate delineation.

It is quite difficult to carry out a validation of the river valley bottom since there is no measurable quantity (Gallant and Dowling, 2003). We, therefore, opted to use the GDE as a proxy of finding if our delineated river valley bottom is in agreement

with nature types commonly found within these areas. Our validation revealed that more than 55% and in one case 67% (Table 2 and Figure 11) of our delineated river valley bottoms contain GDE. We expected this since farmers have been reclaiming the low-lying areas in Denmark during the last 200 years before the first map was created. We believe that the validation would have given better results if the first mapping was conducted before the reclamation of these areas.

The tool performs well but has some limitations. The first limitation applies to the coastal areas where the delineation

picks up strips of the beach due to their low elevation (Figure 12a). Another limitation is in very wide laterally unconfined areas such as fluvial plains in downstream coastal areas. Due to the size of the floodplain, the tool sometimes fails to delineate the entire low-lying area and resolves to delineate valleys from individual rivers (Figure 12b). There are also limitations concerning input data such as only being able to delineate a river valley bottom in areas that have river data. It is therefore important to use a good layer of the river system with the right level of detail as input. Also, the DEM resolution should be at

the level of detail of capturing the confined headwater valleys. E.g. if the interest was in capturing valleys that are less than 10 m then a 10 m DEM would be insufficient for the task.

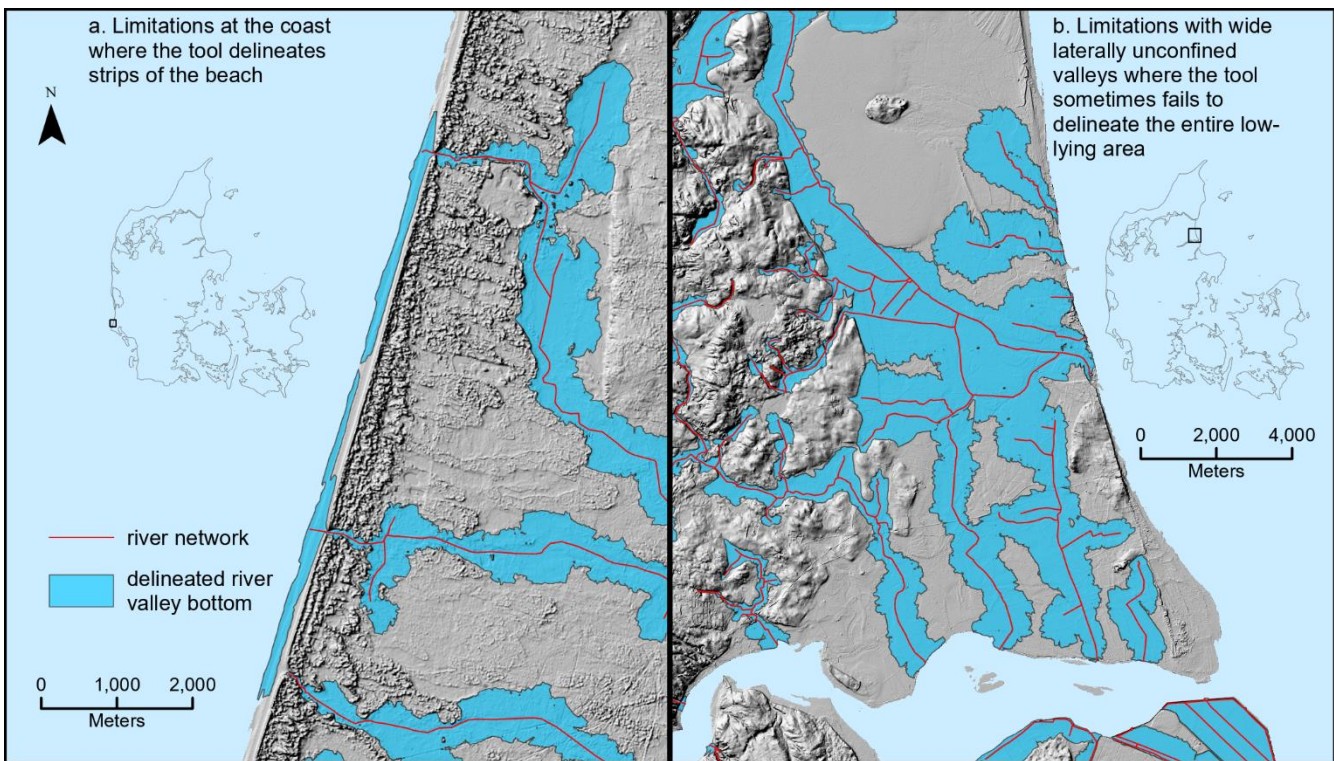

**Figure 12: Limitations of our river valley bottom delineation tool showing (a) limitations at coastal areas in which the tool delineates strips of the beach and (b) limitations with wide laterally unconfined valleys where in some cases the tool fails to delineate the entire low-lying area delineating valleys from individual rivers only.**

## 6 Conclusion

This study presents a new GIS tool for the delineation of the river valley bottom across multiple drainage networks. This is an important management tool since it can delineate an area that has been the focus of management actions in terms of protecting rivers against upland nutrient pollution. We use novel, automatic GIS techniques and illustrate the development of the tool through steps using the Gudenå drainage basin in Denmark. The main method is cost distance accumulation using a river network as the source and slope derived from a Digital Elevation Model (DEM) as cost. We then find a threshold river valley bottom boundary using wetland areas adjacent to rivers as a guide. We run these steps for all drainage basins of Denmark eventually creating a national map of the river valley bottom. The tool visually performs well by extracting valley bottoms that appear to be within the lowest areas of the terrain adjacent to the river network. We validate the resulting tool by finding an overlap of the delineated river valley bottom with Groundwater Dependent Ecosystems (GDE) at selected areas. The validation reveals that at least half to two-thirds of the river valley bottom contains GDE, which is what we expected since the GDE data we used to validate was collected at a historical time during which farmers had already started reclamation low-lying areas. However, the tool has limitations such as delineating beach areas in river coastal outlets and sometimes fails to delineate the





whole low-lying area in wide unconfined valleys. Additionally, the tool will only delineate a river valley bottom at an area that contains river data, and therefore a good river network with the right level of detail should be used as input. We expect that the resulting map can be used for planning and policy support in terms of managing the economic and sustainable use of river valley bottom areas in Denmark e.g. agriculture.

**Code and data availability**

Code and data are available from the authors upon request.

**Author contributions**

GLS, MHG, BVI, BN, and CDB conceptualized the project. GLS performed the analysis and wrote the first draft. GLS, MHG,
BVI, and BN discussed the results. GLS, MHG, BVI, BN, MBG reviewed and edited the paper. MHG, BVI, and BN supervised the project.

**Competing interests**

The authors declare that they have no conflict of interest.

**Acknowledgments**

This work was funded by the Graduate School of Science and Technology, Aarhus University.

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
