# Peer review of "A stepwise GIS approach for the delineation of river valley bottom within drainage basins using a cost distance accumulation analysis"

_Hydrology and Earth System Sciences, 2020_

## Referee Comment (RC1) · Anonymous Referee #1 · 10 Aug 2020

Gasper at al. outline a method to delineate flood plain riparian areas based on a least cost algorithm which uses "Terrain slope" as cost factor to minimize. I like the idea, it is nice to see how it was applied to Denmark (though the reference data seemed a bit coarse) and how some of the key limitations where highlighted (including e.g., use in coastal areas, which is something I really did not have on the radar also it makes very much sense). However, and believe me that my heart truly feels heavy saying this, the concept already exists and was presented and tested by Murphy et al. (2009), further applied by White et. al. in 2012 in Canada, followed by a Swedish case study in 2014 (Ågren et al 2014) and tested on large scale and in combination with machine learning (again in Sweden) by Lidberg et. al. (2020). (Disclaimer: I have actually *not* been

involved in any of these studies).

I have tried to come up with ideas on how to save this manuscript but since its core is the already published method I have to recommend the editor to reject it and the authors to (perhaps?) try to use the method in a different context/ frame (perhaps with a stronger focus on it's practical application in a Danish context?).

Good luck!

Murphy, P. N. C., Ogilvie, J., & Arp, P. (2009). Topographic modelling of soil moisture conditions: a comparison and verification of two models. European Journal of Soil Science, 60(1), 94–109. https://doi.org/10.1111/j.1365-2389.2008.01094.x White, B., Ogilvie, J., Campbell, D. M. H. M. H., Hiltz, D., Gauthier, B., Chisholm, H. K. H., et al. (2012). Using the Cartographic Depth-to-Water Index to Locate Small Streams and Associated Wet Areas across Landscapes. Canadian Water Resources Journal / Revue Canadienne Des Ressources Hydriques, 37(4), 333–347. https://doi.org/10.4296/cwrj2011-909 Ågren, A. M., Lidberg, W., Strömgren, M., Ogilvie, J., & Arp, P. A. (2014). Evaluating digital terrain indices for soil wetness mapping – a Swedish case study. Hydrol. Earth Syst. Sci., 18(9), 3623–3634. https://doi.org/10.5194/hess-18-3623-2014 Lidberg, W., Nilsson, M., & Ågren, A. (2020). Using machine learning to generate high-resolution wet area maps for planning forest management: A study in a boreal forest landscape. Ambio, 49(2), 475–486. https://doi.org/10.1007/s13280-019-01196-9

---

## Author Comment (AC1) · 18 Aug 2020

Dear Anonymous Referee #1,

Thank you for your feedback and concerns. We have reviewed the DTW model, and although similar, it has a difference in the underlying computation. DTW model computes the accumulative slope values along a least-cost path by summing up the slopes within cells. Our method uses the ArcGIS function Cost Distance Accumulation which computes the accumulated cost distance by considering nodes (see reference). The cost distance accumulation is the sum of an average of the slopes taken at the nodes when moving between cells. This implies that results from our method should have a

smoother accumulative effect than those from the DTW model. We developed a practical tool that can easily be used with the ArcGIS Software. We can however reference the DTW model for the practical similarity.

Reference: https://pro.arcgis.com/en/pro-app/tool-reference/spatial-analyst/how-the-cost-distance-tools-work.htm

---

## Referee Comment (RC2) · Anonymous Referee #2 · 17 Sep 2020

The manuscript proposes a tool for river valley bottom delineation. The topic is surely interesting and the manuscript easy to read, however in the manuscript there are some drawbacks listed as in the following, that should be addressed.

1) In the Introduction the review is not complete either concerning the existing flooding based methods (for instance: Nardi et al. 2019) and concerning the slope-based approaches (for instance: DTW approach, as mentioned by the first reviewer).

2) It should be clearer the added value of the proposed tool comparing the results to the other available methods.

3) I found unclear the difference between floodplain and river valley bottom definition.

[Figure]

4) It should be clarified the proposed method description. If the reader would like to apply the method and he/she follows the Section 3.2, I do not think he/she would be able to do that. For instance line 171 "small positive value" is vague; the same for lines 186-187; and 193-194.

5) why the validation is not performed on the entire Denmark? three validation areas can not support conclusion for the reasons mentioned by the authors in the discussion.

References:

Nardi, F., Annis, A., Baldassarre, G.D., Vivoni, E.R., Grimaldi, S. GFPLAIN250m, a global high-resolution dataset of earth's floodplains (2019) Scientific Data, 6, art. no. 309.

---

## Author Comment (AC2) · 2 Oct 2020

Thank you for your feedback and concerns. We have addressed them as follows:

1) In the Introduction the review is not complete either concerning the existing flooding based methods (for instance: Nardi et al. 2019) and concerning the slope-based approaches (for instance: DTW approach, as mentioned by the first reviewer).

- As we responded to the first reviewer, we can reference the DTW model for its practical similarity to our model. The tools we presented are some that exist, Nardi et al. 2019 is a global framework for floodplain delineation, which may not be suitable for

local delineations.

2) It should be clearer the added value of the proposed tool comparing the results to the other available methods.

- We explained the value of this work in the discussion section, where we outline that our tool solves issues of scaling through the cost distance accumulation algorithm by resolving cutoff valley bottom boundaries at relative distances in different configurations of valleys within the same drainage basin.

3) I found unclear the difference between floodplain and river valley bottom definition.

- As we explained in the paper, the valley bottom includes the river and its floodplain (line 41). However, we will change the description of the definition of the differences between floodplain and river valley bottom in order to make it more clear, that the floodplain is the area adjacent to a river i.e., stretching from the riverbank to the edge of the valley.

4) It should be clarified the proposed method description. If the reader would like to apply the method and he/she follows the Section 3.2, I do not think he/she would be able to do that. For instance line 171 "small positive value" is vague; the same for lines 186-187; and 193-194.

- Line 171: The "small positive value" could be construed as vague, for clarity, we can change that to "a value close to zero but not zero e.g., 0.0000000001".

- Line 186-187: I believe this makes sense since we are explaining that our delineation uses wetland areas that are adjacent to rivers as a guideline.

- Line 193-194: Also makes sense since we follow up with Figure 8b and Figure 9, explaining why we use 500.

5) why the validation is not performed on the entire Denmark? three validation areas can not support conclusion for the reasons mentioned by the authors in the discussion.

- Unfortunately, these three areas are the only ones that we could use to validate the model, it is the only measured data that have. It is a monumental task to digitize old topographical maps on a national scale even in Denmark

———————————————